

# Entropic analysis of optomechanical entanglement for a nanomechanical resonator coupled to an optical cavity field

**Jeong Ryeol Choi**[⋆]

Department of Nanoengineering, College of Convergence and Integrated Science,
Kyonggi University, Yeongtong-gu, Suwon, Gyeonggi-do 16227, Republic of Korea

⋆ choiardor@hanmail.net

## Abstract

We investigate entanglement dynamics for a nanomechanical resonator coupled to an optical cavity field through the analysis of the associated entanglement entropies. The effects of time variation of several parameters, such as the optical frequency and the coupling strength, on the evolution of entanglement entropies are analyzed. We consider three kinds of entanglement entropies as the measures of the entanglement of subsystems, which are the linear entropy, the von Neumann entropy, and the Rényi entropy. The analytic formulae of these entropies are derived in a rigorous way using wave functions of the system. In particular, we focus on time behaviors of entanglement entropies in the case where the optical frequency is modulated by a small oscillating factor. We show that the entanglement entropies emerge and increase as the coupling strength grows from zero. The entanglement entropies fluctuate depending on the adiabatic variation of the parameters and such fluctuations are significant especially in the strong coupling regime. Our research may deepen the understanding of the optomechanical entanglement, which is crucial in realizing hybrid quantum-information protocols in quantum computation, quantum networks, and other domains in quantum science.

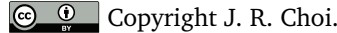

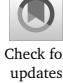

# 1   Introduction

Many novel quantum phenomena and related effects are important in realizing next generation quantum technologies. Most of such phenomena can be produced by coupling nano- and micromechanical oscillators with a variety of other systems, such as electrons [1], photons [2], qubits [3], and magnetic devices [4]. In particular, photonic couplings provide a powerful platform for state-of-the-art optomechanical techniques applicable to signal routing and protection [5,6], control of phononic structures [7], producing slow/fast light [8], and chiral cooling [9]. Besides, coupled optomechanical systems are promising candidates for highly sensitive quantum devices required in quantum information processes and quantum state tomographies.

The entanglement of mechanical modes with cavity fields through the effect of radiation pressure [10] provides a key paradigm for precision measurements in quantum metrology. Such entanglement can be employed, for example, to construct information networks which connect flying bits with solid bits [11,12], and to store quantum information [13]. Hence, the generation of quantum entanglement and its control are very important in optomechanics. Fundamental quantum properties associated with nonclassical correlations, quantum coherence, and decoherence can also be understood from entanglement dynamics.

It is possible to generate steady-state entanglement in optomechanical systems incorporated with mechanical oscillators by adjusting the effective frequency of a movable mirror [14, 15]. Although optomechanical systems serve as good coupling mediums for entanglement [9, 16], the mechanism related to such entanglement is not fully understood yet. We should resolve this problem, together with the lack of the knowledge for determining parameters for wide range of practical applications of them.

If we think of the fact that maximally entangled states often serve as optimal inputs in the quantum-information protocols [17], a rigorous quantification of such an entanglement is highly required. The entanglement can be quantified by several measures, such as entanglement entropies [18–21], the logarithmic negativity [14], the generalized concurrence [22], or Gaussian-type basis functions [23]. Among those, we are interested in entanglement entropies in this work, which are deeply related to entanglement properties of a state. Entanglement entropies provide a bedrock concept of entanglement on the basis of quantum statistical mechanics, quantifying how strongly the subsystems are entangled.

The purpose of this work is to quantify the entanglement of an optomechanical system from a strict mathematical framework of the entanglement entropies. The system that we consider is a coupling of a nanomechanical resonator and a cavity field, where parameters of the system vary adiabatically in time. In particular, we see the effects of a small sinusoidal variation of the frequency of the optical oscillator on the entanglement entropies. The time behavior of the entanglement entropies in that case will be analyzed from various angles.

Organization of this article is as follows. In Sec. 2, we will show how to describe the optomechanical system that we manage based on Hamiltonian dynamics. The quantum solutions of the system will be derived in Sec. 3 using the unitary transformation method, which is a useful mathematical tool for treating time-dependent Hamiltonian systems [24, 25]. Due to time variation of the parameters, our system is described by a time-dependent Hamiltonian.

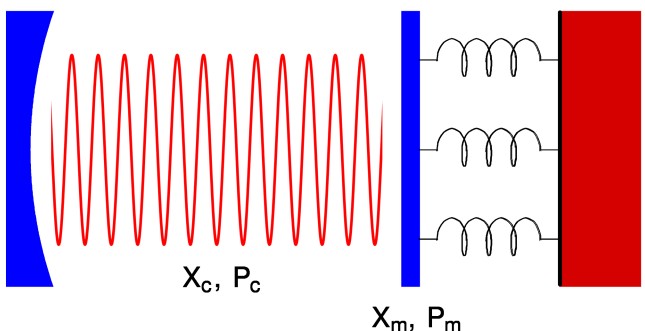

Figure 1: Schematic of the nanomechanical resonator coupled to a cavity field.

Based on such quantum solutions, we will derive a reduced density matrix in Sec. 4. Several types of entanglement entropies will be evaluated by taking advantage of the reduced density matrix. The linear entropy will be investigated at first in Sec. 5 separately for mechanical and optical parts of the system. And then, we will extend our development to the von Neumann entropy and the Rényi entropy in Sec. 6. As well as the characteristics of such entanglement entropies, the differences and similarities between them will be analyzed. Concluding remarks are given in the last section.

## 2 Preliminary description of the optomechanical system

We consider a nanoresonator coupled to a cavity field via a time-dependent coupling strength $g(t)$, whereas the cavity field is driven by a laser field of which frequency is $\omega_L$ (see Fig. 1). The nanomechanical resonator interacts with the cavity field through the radiation-pressure force carried by the momentum of light. We assume that, as well as $g$, the frequency $\Delta$ of the optical oscillator and the frequency $\omega_m$ of the mechanical resonator exhibit possible variations in time. The consideration of such time variations of the parameters is the difference of our description of the system from that of Ref. [2], which corresponds to the case that all parameters are *in*dependent of time. We further suppose that the time variations of the three parameters, $g(t)$, $\Delta(t)$, and $\omega_m(t)$, are sufficiently slow so that the evolution of the coupled system satisfies the adiabatic condition.

If we consider that cavity is driven by a laser field, the relation between the optical frequency $\Delta$ and the cavity frequency $\omega_c$ is given by $\Delta = \omega_c - \omega_L - \delta_{rp}$, where $\delta_{rp}$ is the shift of the cavity frequency by radiation pressure [2]. On the other hand, the coupling strength is given by $g(t) = G(t)\sqrt{\langle n_c \rangle}$, where $G(t) = [\omega_c(t)/L(t)]\sqrt{\hbar/[m\omega_m(t)]}$, $m$ is effective mass of the resonator, $L$ is the cavity length, and $\langle n_c \rangle$ is the mean cavity photon number [2].

Strong coupling in optomechanical systems is favorable for preparing entangled states or squeezed entangled states, because it is difficult to obtain steady-state entanglement if the interaction between subsystems is too weak [2,15]. Furthermore, the effects of decoherence in coupled systems can be overcome or at least reduced, if we adopt a strong coupling. Hence, the strong coupling, fortified by these advantages, enables us to carry out quantum experiments with proper control of mechanical quantum states. [2].

From standard mean-field expansion of the interaction, we have a coupled harmonic oscillator description of the system in terms of the dimensionless mechanical position operator $\hat{X}_m$ and the dimensionless optical quadrature operator $\hat{X}_c$. These two operators are given in the form

$$\hat{X}_m = \frac{1}{\sqrt{2}}(\hat{a}_m + \hat{a}_m^\dagger) \qquad \hat{X}_c = \frac{1}{\sqrt{2}}(\hat{a}_c + \hat{a}_c^\dagger), \tag{1}$$

where $\hat{a}_{\mathrm{m}}$ ($\hat{a}_{\mathrm{c}}$) and $\hat{a}_{\mathrm{m}}^{\dagger}$ ($\hat{a}_{\mathrm{c}}^{\dagger}$) are annihilation and creation operators, respectively, for the mechanical mode (optical mode). Their canonical conjugate operators are given, respectively, by $\hat{P}_{\mathrm{m}} = -i\partial/\partial X_{\mathrm{m}}$ and $\hat{P}_{\mathrm{c}} = -i\partial/\partial X_{\mathrm{c}}$. These operators can also be represented as

$$\hat{P}_{\mathrm{m}} = \frac{i}{\sqrt{2}}(\hat{a}_{\mathrm{m}}^{\dagger} - \hat{a}_{\mathrm{m}}) \qquad \hat{P}_{\mathrm{c}} = \frac{i}{\sqrt{2}}(\hat{a}_{\mathrm{c}}^{\dagger} - \hat{a}_{\mathrm{c}}). \tag{2}$$

We confirm that the canonical operators satisfy the commutation relations, $[\hat{X}_{\mathrm{m}}, \hat{P}_{\mathrm{m}}] = [\hat{X}_{\mathrm{c}}, \hat{P}_{\mathrm{c}}] = i$.

The linearized Hamiltonian for the optomechanical system that we have considered in the time-dependent regime is now expressed to be

$$\hat{H} = \frac{\hbar\omega_{\mathrm{m}}(t)}{2}(\hat{X}_{\mathrm{m}}^2 + \hat{P}_{\mathrm{m}}^2) + \frac{\hbar\Delta(t)}{2}(\hat{X}_{\mathrm{c}}^2 + \hat{P}_{\mathrm{c}}^2) - \hbar g(t)\hat{X}_{\mathrm{m}}\hat{X}_{\mathrm{c}}. \tag{3}$$

For the case in which the parameters, $\Delta$, $\omega_{\mathrm{m}}$, and $g$, are constants, this Hamiltonian reduces to that of Ref. [2].

Quantum solutions of the system can be derived through diagonalization of the Hamiltonian. For a coupled system, we may need to consider the unitary transformation or the canonical transformation approach for its exact diagonalization. However, conventionally, many authors adopt some approximations in such a case, considering a Hamiltonian in a frame of simply rotated coordinates. We confirm that such a rotation is described as

$$\begin{pmatrix} \hat{X}_{\mathrm{m}} \\ \hat{X}_{\mathrm{c}} \\ \hat{P}_{\mathrm{m}} \\ \hat{P}_{\mathrm{c}} \end{pmatrix} = \begin{pmatrix} \cos\theta(t) & \sin\theta(t) & 0 & 0 \\ -\sin\theta(t) & \cos\theta(t) & 0 & 0 \\ 0 & 0 & \cos\theta(t) & \sin\theta(t) \\ 0 & 0 & -\sin\theta(t) & \cos\theta(t) \end{pmatrix} \begin{pmatrix} \hat{X}_{+} \\ \hat{X}_{-} \\ \hat{P}_{+} \\ \hat{P}_{-} \end{pmatrix}, \tag{4}$$

where $\hat{X}_{\pm}$ and $\hat{P}_{\pm}$ are canonical operators in the rotated frame. If we take $\theta(t)$ in the above equation in the form

$$\theta(t) = \frac{1}{2}\mathrm{atan}(\omega_{\mathrm{m}}(t) - \Delta(t), 2g(t)), \tag{5}$$

where $\vartheta \equiv \mathrm{atan}(z_1, z_2)$ is the two-variables inverse function of $\tan\vartheta = z_2/z_1$, the Hamiltonian, Eq. (3), is represented to be

$$\hat{H} = \hat{H}_{+} + \hat{H}_{-} + \hat{h}, \tag{6}$$

where

$$\hat{H}_{\pm} = \frac{\hbar}{2}[\omega_{X,\pm}(t)\hat{X}_{\pm}^2 + \omega_{P,\pm}(t)\hat{P}_{\pm}^2], \tag{7}$$

$$\hat{h} = \frac{\hbar}{2}[\omega_{\mathrm{m}}(t) - \Delta(t)]\sin[2\theta(t)]\hat{P}_{+}\hat{P}_{-}, \tag{8}$$

with

$$\omega_{X,+} = \omega_{\mathrm{m}}(t)\cos^2\theta(t) + \Delta(t)\sin^2\theta(t) + g(t)\sin[2\theta(t)], \tag{9}$$

$$\omega_{X,-} = \omega_{\mathrm{m}}(t)\sin^2\theta(t) + \Delta(t)\cos^2\theta(t) - g(t)\sin[2\theta(t)], \tag{10}$$

$$\omega_{P,+} = \omega_{\mathrm{m}}(t)\cos^2\theta(t) + \Delta(t)\sin^2\theta(t), \tag{11}$$

$$\omega_{P,-} = \omega_{\mathrm{m}}(t)\sin^2\theta(t) + \Delta(t)\cos^2\theta(t). \tag{12}$$

From this procedure, we have eliminated the cross term that involves $\hat{X}_{\mathrm{m}}\hat{X}_{\mathrm{c}}$ in the original Hamiltonian. However, a new cross term described by $\hat{P}_{+}\hat{P}_{-}$ has appeared. We can easily confirm from Eq. (8) that this new term disappears in the case of resonance ($\omega_{\mathrm{m}}(t) = \Delta(t)$). For this reason, some authors manage the system considering only the resonance case in order to avoid mathematical complexity (see, for example, Refs. [2, 20, 21, 26, 27]).

# 3 Unitary transformation and wave functions

For a general treatment of the system including non-resonance cases, a rigorous mathematical procedure beyond the simple rotation method may be necessary. We use the unitary transformation method [24,25] in order to meet this demand. The intricate original Hamiltonian can be transformed to a simple form by means of a unitary transformation. Then the Schrödinger solutions in the transformed system may be easily obtained due to the simplicity of the transformed Hamiltonian. From the inverse transformation of such solutions using the same unitary operator, it is also possible to obtain the Schrödinger solutions in the original system without difficulty. This is the main idea of the unitary transformation method, which is incorporated to the purpose of deriving complete quantum solutions. The effect of the quantum properties resulting from entanglement of the wave packets may also be better envisioned by adopting this alternative approach.

We now introduce a unitary operator as

$$\hat{U} = \exp\{-i\varphi(t)[\beta(t)\hat{P}_\mathrm{m}\hat{X}_\mathrm{c} - \beta^{-1}(t)\hat{P}_\mathrm{c}\hat{X}_\mathrm{m}]\}, \tag{13}$$

where

$$\varphi = \frac{1}{2}\mathrm{atan}(\beta\omega_\mathrm{m}(t) - \beta^{-1}\Delta(t), 2g(t)), \tag{14}$$

$$\beta = \sqrt{\frac{\omega_\mathrm{m}(t)}{\Delta(t)}}. \tag{15}$$

From the transformation of the original Hamiltonian by means of $\hat{U}$ using the relation

$$\hat{\mathcal{H}} = \hat{U}^{-1}\hat{H}\hat{U} - i\hbar\hat{U}^{-1}\frac{\partial\hat{U}}{\partial t}, \tag{16}$$

we can have the Hamiltonian $\hat{\mathcal{H}}$ associated to the transformed system. A straightforward evaluation of the above equation gives

$$\begin{aligned}
\hat{\mathcal{H}} &= \frac{\hbar}{2}[\omega_{X,\mathrm{m}}(t)\hat{X}_\mathrm{m}^2 + \omega_\mathrm{m}(t)\hat{P}_\mathrm{m}^2 + \omega_{X,\mathrm{c}}(t)\hat{X}_\mathrm{c}^2 + \Delta(t)\hat{P}_\mathrm{c}^2] \\
&\quad - \hbar[\dot{\varphi}_1(t)\hat{P}_\mathrm{m}\hat{X}_\mathrm{c} - \dot{\varphi}_2(t)\hat{P}_\mathrm{c}\hat{X}_\mathrm{m}],
\end{aligned} \tag{17}$$

where $\varphi_1(t) = \varphi(t)\beta(t)$, $\varphi_2(t) = \varphi(t)\beta^{-1}(t)$, and

$$\omega_{X,\mathrm{m}} = \omega_\mathrm{m}(t)\cos^2\varphi(t) + \Delta(t)\beta^{-2}(t)\sin^2\varphi(t) + g(t)\beta^{-1}(t)\sin[2\varphi(t)], \tag{18}$$

$$\omega_{X,\mathrm{c}} = \omega_\mathrm{m}(t)\beta^2(t)\sin^2\varphi(t) + \Delta(t)\cos^2\varphi(t) - g(t)\beta(t)\sin[2\varphi(t)]. \tag{19}$$

Let us assume that the variations of $\varphi_1(t)$ and $\varphi_2(t)$ over time are sufficiently slow. This assumption is actually equivalent to the previous assumption that the variations of $g(t)$, $\Delta(t)$, and $\omega_\mathrm{m}(t)$ are slow. Then it is possible to neglect the last term that involves $\dot{\varphi}_1(t)$ and $\dot{\varphi}_2(t)$ in Eq. (17), leading to a simple diagonalized Hamiltonian:

$$\hat{\mathcal{H}} = \frac{\hbar}{2}[\omega_{X,\mathrm{m}}(t)\hat{X}_\mathrm{m}^2 + \omega_\mathrm{m}(t)\hat{P}_\mathrm{m}^2 + \omega_{X,\mathrm{c}}(t)\hat{X}_\mathrm{c}^2 + \Delta(t)\hat{P}_\mathrm{c}^2]. \tag{20}$$

The manage of the system based on this Hamiltonian may be much easier from quantum-mechanical point of view than the use of the original Hamiltonian. By the way, if we choose $\beta = 1$ instead of Eq. (15), Eq. (14) reduces to $\theta(t)$ given in Eq. (5). This means that the previous simple rotation procedure is available only for the resonance case, as expected.

The characterization of quantum properties of a (composite) system starts from the explicit mathematical formulation of the wave functions. The mysterious property of nonlocality associated with quantum entanglement may also be understandable through the wave functions from the most fundamental level. If we write the Schrödinger equation for the simple but time-dependent Hamiltonian $\hat{\mathcal{H}}$ as

$$i\hbar\frac{\partial\Psi_{n,l}(X_{\mathrm{m}},X_{\mathrm{c}},t)}{\partial t} = \hat{\mathcal{H}}\Psi_{n,l}(X_{\mathrm{m}},X_{\mathrm{c}},t), \tag{21}$$

the overall wave functions in the transformed system are a linear product of the mechanical and optical parts of the wave functions, such that

$$\Psi_{n,l}(X_{\mathrm{m}},X_{\mathrm{c}},t) = \Psi_n(X_{\mathrm{m}},t)\tilde{\Psi}_l(X_{\mathrm{c}},t). \tag{22}$$

If we regard Eq. (20), each component in the above equation is of the form (see Appendix A)

$$\Psi_n(X_{\mathrm{m}},t) = \Phi_n(X_{\mathrm{m}},t)\exp[i\delta_n(t)], \tag{23}$$
$$\tilde{\Psi}_l(X_{\mathrm{c}},t) = \tilde{\Phi}_l(X_{\mathrm{c}},t)\exp[i\tilde{\delta}_l(t)], \tag{24}$$

where $\delta_n(t)$ and $\tilde{\delta}_l(t)$ are phases. Here, $\Phi_n(X_{\mathrm{m}},t)$ and $\delta_n(t)$ ($\tilde{\Phi}_l(X_{\mathrm{c}},t)$ and $\tilde{\delta}_l(t)$) are given in terms of $\eta_{\mathrm{m}}(t)$ and $\gamma_{\mathrm{m}}(t)$ ($\eta_{\mathrm{c}}(t)$ and $\gamma_{\mathrm{c}}(t)$) which are time-dependent factors of the classical solution given in Eq. (A.13) (Eq. (A.14)) in Appendix A, such that

$$\Phi_n(X_{\mathrm{m}},t) = \sqrt[4]{\frac{\dot{\gamma}_{\mathrm{m}}(t)}{\pi\omega_{\mathrm{m}}(t)}}\frac{1}{\sqrt{2^n n!}}H_n\left(\sqrt{\frac{\dot{\gamma}_{\mathrm{m}}(t)}{\omega_{\mathrm{m}}(t)}}X_{\mathrm{m}}\right)\exp\left[-Y_{\mathrm{m}}(t)X_{\mathrm{m}}^2\right], \tag{25}$$
$$\delta_n(t) = -(n+1/2)\gamma_{\mathrm{m}}(t), \tag{26}$$

$$\tilde{\Phi}_l(X_{\mathrm{c}},t) = \sqrt[4]{\frac{\dot{\gamma}_{\mathrm{c}}(t)}{\pi\Delta(t)}}\frac{1}{\sqrt{2^l l!}}H_l\left(\sqrt{\frac{\dot{\gamma}_{\mathrm{c}}(t)}{\Delta(t)}}X_{\mathrm{c}}\right)\exp\left[-Y_{\mathrm{c}}(t)X_{\mathrm{c}}^2\right], \tag{27}$$
$$\tilde{\delta}_l(t) = -(l+1/2)\gamma_{\mathrm{c}}(t), \tag{28}$$

where $H_n$ are $n$th order Hermite polynomials, while

$$Y_{\mathrm{m}}(t) = \frac{1}{2\omega_{\mathrm{m}}(t)}\left(\dot{\gamma}_{\mathrm{m}}(t)-i\frac{\dot{\eta}_{\mathrm{m}}(t)}{\eta_{\mathrm{m}}(t)}\right), \tag{29}$$
$$Y_{\mathrm{c}}(t) = \frac{1}{2\Delta(t)}\left(\dot{\gamma}_{\mathrm{c}}(t)-i\frac{\dot{\eta}_{\mathrm{c}}(t)}{\eta_{\mathrm{c}}(t)}\right). \tag{30}$$

The wave functions $\psi_{n,l}(X_{\mathrm{m}},X_{\mathrm{c}},t)$ in the original system (untransformed system) can be obtained from the inverse unitary transformation:

$$\psi_{n,l}(X_{\mathrm{m}},X_{\mathrm{c}},t) = \hat{U}\Psi_{n,l}(X_{\mathrm{m}},X_{\mathrm{c}},t). \tag{31}$$

From a minor computation using Eq. (13) and Eq. (22) with subsequent equations, we have

$$\psi_{n,l}(X_{\mathrm{m}},X_{\mathrm{c}},t) = \psi_n(X_{\mathrm{m}},X_{\mathrm{c}},t)\tilde{\psi}_l(X_{\mathrm{m}},X_{\mathrm{c}},t), \tag{32}$$

where

$$\psi_n(X_{\mathrm{m}},X_{\mathrm{c}},t) = \phi_n(X_{\mathrm{m}},X_{\mathrm{c}},t)\exp[i\delta_n(t)], \tag{33}$$
$$\tilde{\psi}_l(X_{\mathrm{m}},X_{\mathrm{c}},t) = \tilde{\phi}_l(X_{\mathrm{m}},X_{\mathrm{c}},t)\exp[i\tilde{\delta}_l(t)], \tag{34}$$

with

$$\phi_n(X_\mathrm{m},X_\mathrm{c},t) = \sqrt[4]{\frac{\dot{\gamma}_\mathrm{m}(t)}{\pi\omega_\mathrm{m}(t)}}\frac{1}{\sqrt{2^n n!}}H_n\left(\sqrt{\frac{\dot{\gamma}_\mathrm{m}(t)}{\omega_\mathrm{m}(t)}}\mathcal{X}_\mathrm{m}\right)\exp\left[-Y_\mathrm{m}(t)\mathcal{X}_\mathrm{m}^2\right], \tag{35}$$

$$\tilde{\phi}_l(X_\mathrm{m},X_\mathrm{c},t) = \sqrt[4]{\frac{\dot{\gamma}_\mathrm{c}(t)}{\pi\Delta(t)}}\frac{1}{\sqrt{2^l l!}}H_l\left(\sqrt{\frac{\dot{\gamma}_\mathrm{c}(t)}{\Delta(t)}}\mathcal{X}_\mathrm{c}\right)\exp\left[-Y_\mathrm{c}(t)\mathcal{X}_\mathrm{c}^2\right], \tag{36}$$

while

$$\mathcal{X}_\mathrm{m} = X_\mathrm{m}\cos\varphi(t) - \beta(t)X_\mathrm{c}\sin\varphi(t), \tag{37}$$

$$\mathcal{X}_\mathrm{c} = X_\mathrm{c}\cos\varphi(t) + \beta^{-1}(t)X_\mathrm{m}\sin\varphi(t). \tag{38}$$

Thus, using Eq. (32) with Eqs. (33)-(38) and Eqs. (26) and (28), we can represent the full wave function in the form

$$\psi(X_\mathrm{m},X_\mathrm{c},t) = \sum_{n=0}^{\infty}\sum_{l=0}^{\infty}c_{n,l}\psi_{n,l}(X_\mathrm{m},X_\mathrm{c},t), \tag{39}$$

where $c_{n,l}$ are complex numbers which obey the condition $\sum_{n=0}^{\infty}\sum_{l=0}^{\infty}|c_{n,l}|^2 = 1$. In the subsequent sections, the wave functions in the original system developed here will be used in analyzing the entanglement properties of the system through the quantification of the entanglement entropies.

# 4 Reduced density matrix

It is possible to derive the analytical formula of the entanglement entropies by using the reduced density matrix. Let us see the reduced density matrix before the main development of the entanglement structure of the system. The reduced density matrices in the Fock state are easily calculated from the density matrices which are given by

$$\rho_{n,l}(X_\mathrm{m},X_\mathrm{c},X_\mathrm{m}',X_\mathrm{c}',t) = \psi_{n,l}^*(X_\mathrm{m}',X_\mathrm{c}',t)\psi_{n,l}(X_\mathrm{m},X_\mathrm{c},t). \tag{40}$$

According to fundamental quantum mechanics, we can quantitatively describe mixed states in optomechanical systems as well as the pure state through the use of these matrices.

Preparing the ground state for the mechanical resonator using the technique of optical laser cooling has been a goal of continuing research [11,21,28,29]. Ground-state cooling provides a new route for exploring the quantum regime of mechanical systems with preparation of nonclassical states such as squeezed, entangled, and superposition states. If we regard this, it may be preferable to focus on the reduced density matrix in the ground state.

Reduced ground-state density matrices for mechanical and optical parts of the system are given respectively by [28,30]

$$\rho_{0,0}^\mathrm{R}(X_\mathrm{m},X_\mathrm{m}',t) = \int_{-\infty}^{\infty}\psi_{0,0}^*(X_\mathrm{m}',X_\mathrm{c},t)\psi_{0,0}(X_\mathrm{m},X_\mathrm{c},t)dX_\mathrm{c}, \tag{41}$$

$$\tilde{\rho}_{0,0}^\mathrm{R}(X_\mathrm{c},X_\mathrm{c}',t) = \int_{-\infty}^{\infty}\psi_{0,0}^*(X_\mathrm{m},X_\mathrm{c}',t)\psi_{0,0}(X_\mathrm{m},X_\mathrm{c},t)dX_\mathrm{m}. \tag{42}$$

Let us first see for the mechanical part. A straightforward evaluation of the integration given in Eq. (41) using Eq. (32) with $n = l = 0$ results in

$$\rho_{0,0}^\mathrm{R}(X_\mathrm{m},X_\mathrm{m}',t) = \sqrt{\frac{N_\mathrm{m}}{\pi}}\exp\{-[A_\mathrm{m}X_\mathrm{m}^2 + A_\mathrm{m}^*X_\mathrm{m}'^2] + B_\mathrm{m}X_\mathrm{m}X_\mathrm{m}'\}, \tag{43}$$

where

$$
\begin{aligned}
A_{\mathrm{m}} = & \frac{1}{\beta^2 W_{\mathrm{m}}} \{ (Y_{\mathrm{c}} + Y_{\mathrm{c}}^*) Y_{\mathrm{m}} \beta^2 \cos^4 \varphi + Y_{\mathrm{c}} (Y_{\mathrm{m}} + Y_{\mathrm{m}}^*) \beta^2 \sin^4 \varphi \\
& + [Y_{\mathrm{m}} Y_{\mathrm{m}}^* \beta^4 + Y_{\mathrm{c}} (Y_{\mathrm{c}}^* + 2 Y_{\mathrm{m}} \beta^2)] \cos^2 \varphi \sin^2 \varphi \},
\end{aligned} \tag{44}
$$

$$
B_{\mathrm{m}} = \frac{1}{2 W_{\mathrm{m}}} [Y_{\mathrm{m}} \beta - Y_{\mathrm{c}}/\beta][Y_{\mathrm{m}}^* \beta - Y_{\mathrm{c}}^*/\beta] \sin^2(2\varphi), \tag{45}
$$

$$
N_{\mathrm{m}} = 2\mathrm{Re}[A_{\mathrm{m}}] - B_{\mathrm{m}} = \frac{\dot{\gamma}_{\mathrm{m}} \dot{\gamma}_{\mathrm{c}}}{W_{\mathrm{m}} \omega_{\mathrm{m}} \Delta}, \tag{46}
$$

$$
\begin{aligned}
W_{\mathrm{m}} & = (Y_{\mathrm{c}} + Y_{\mathrm{c}}^*) \cos^2 \varphi + (Y_{\mathrm{m}} + Y_{\mathrm{m}}^*) \beta^2 \sin^2 \varphi \\
& = \frac{\dot{\gamma}_{\mathrm{c}}}{\Delta} \cos^2 \varphi + \frac{\dot{\gamma}_{\mathrm{m}}}{\omega_{\mathrm{m}}} \beta^2 \sin^2 \varphi.
\end{aligned} \tag{47}
$$

We see that the normalization factor $N_{\mathrm{m}}$ is represented in terms of the real part of $A_{\mathrm{m}}$, which can be written as $\mathrm{Re}[A_{\mathrm{m}}] = \frac{1}{2}(A_{\mathrm{m}} + A_{\mathrm{m}}^*)$. By the way, the imaginary part of $A_{\mathrm{m}}$ is given by $\mathrm{Im}[A_{\mathrm{m}}] = \frac{1}{2i}(A_{\mathrm{m}} - A_{\mathrm{m}}^*)$. Using Eq. (44), we can easily confirm that

$$
\begin{aligned}
\mathrm{Re}[A_{\mathrm{m}}] = & \frac{1}{8\beta^2 W_{\mathrm{m}}} \{ Y_{\mathrm{c}} Y_{\mathrm{c}}^* + Y_{\mathrm{m}} Y_{\mathrm{m}}^* \beta^4 + [3(Y_{\mathrm{c}}^* Y_{\mathrm{m}} + Y_{\mathrm{c}} Y_{\mathrm{m}}^*) + 4(Y_{\mathrm{c}} Y_{\mathrm{m}} + Y_{\mathrm{c}}^* Y_{\mathrm{m}}^*)] \beta^2 \\
& - (Y_{\mathrm{c}} - Y_{\mathrm{m}} \beta^2)(Y_{\mathrm{c}}^* - Y_{\mathrm{m}}^* \beta^2) \cos(4\varphi) \},
\end{aligned} \tag{48}
$$

$$
\mathrm{Im}[A_{\mathrm{m}}] = \frac{1}{2i W_{\mathrm{m}}} [Y_{\mathrm{c}} Y_{\mathrm{m}} - Y_{\mathrm{c}}^* Y_{\mathrm{m}}^* + (Y_{\mathrm{c}}^* Y_{\mathrm{m}} - Y_{\mathrm{c}} Y_{\mathrm{m}}^*) \cos(2\varphi)]. \tag{49}
$$

We also derive the ground-state density matrix for the optical part from a similar evaluation, such that

$$
\tilde{\rho}_{0,0}^{\mathrm{R}}(X_{\mathrm{c}}, X_{\mathrm{c}}', t) = \sqrt{\frac{N_{\mathrm{c}}}{\pi}} \exp\{ -[A_{\mathrm{c}} X_{\mathrm{c}}^2 + A_{\mathrm{c}}^* X_{\mathrm{c}}'^2] + B_{\mathrm{c}} X_{\mathrm{c}} X_{\mathrm{c}}' \}, \tag{50}
$$

where

$$
\begin{aligned}
A_{\mathrm{c}} = & \frac{1}{\beta^2 W_{\mathrm{c}}} \{ (Y_{\mathrm{m}} + Y_{\mathrm{m}}^*) Y_{\mathrm{c}} \beta^2 \cos^4 \varphi + Y_{\mathrm{m}} (Y_{\mathrm{c}} + Y_{\mathrm{c}}^*) \beta^2 \sin^4 \varphi \\
& + [Y_{\mathrm{m}} Y_{\mathrm{m}}^* \beta^4 + Y_{\mathrm{c}} (Y_{\mathrm{c}}^* + 2 Y_{\mathrm{m}} \beta^2)] \cos^2 \varphi \sin^2 \varphi \},
\end{aligned} \tag{51}
$$

$$
B_{\mathrm{c}} = \frac{1}{2 W_{\mathrm{c}}} [Y_{\mathrm{m}} \beta - Y_{\mathrm{c}}/\beta][Y_{\mathrm{m}}^* \beta - Y_{\mathrm{c}}^*/\beta] \sin^2(2\varphi), \tag{52}
$$

$$
N_{\mathrm{c}} = 2\mathrm{Re}[A_{\mathrm{c}}] - B_{\mathrm{c}} = \frac{\dot{\gamma}_{\mathrm{m}} \dot{\gamma}_{\mathrm{c}}}{W_{\mathrm{c}} \omega_{\mathrm{m}} \Delta}, \tag{53}
$$

$$
\begin{aligned}
W_{\mathrm{c}} & = (Y_{\mathrm{m}} + Y_{\mathrm{m}}^*) \cos^2 \varphi + (Y_{\mathrm{c}} + Y_{\mathrm{c}}^*) \beta^{-2} \sin^2 \varphi \\
& = \frac{\dot{\gamma}_{\mathrm{m}}}{\omega_{\mathrm{m}}} \cos^2 \varphi + \frac{\dot{\gamma}_{\mathrm{c}}}{\Delta} \beta^{-2} \sin^2 \varphi.
\end{aligned} \tag{54}
$$

The real and imaginary parts of $A_{\mathrm{c}}$ can also be confirmed to be

$$
\begin{aligned}
\mathrm{Re}[A_{\mathrm{c}}] = & \frac{1}{8\beta^2 W_{\mathrm{c}}} \{ Y_{\mathrm{c}} Y_{\mathrm{c}}^* + Y_{\mathrm{m}} Y_{\mathrm{m}}^* \beta^4 + [3(Y_{\mathrm{c}}^* Y_{\mathrm{m}} + Y_{\mathrm{c}} Y_{\mathrm{m}}^*) + 4(Y_{\mathrm{c}} Y_{\mathrm{m}} + Y_{\mathrm{c}}^* Y_{\mathrm{m}}^*)] \beta^2 \\
& - (Y_{\mathrm{c}} - Y_{\mathrm{m}} \beta^2)(Y_{\mathrm{c}}^* - Y_{\mathrm{m}}^* \beta^2) \cos(4\varphi) \},
\end{aligned} \tag{55}
$$

$$
\mathrm{Im}[A_{\mathrm{c}}] = \frac{1}{2i W_{\mathrm{c}}} [Y_{\mathrm{c}} Y_{\mathrm{m}} - Y_{\mathrm{c}}^* Y_{\mathrm{m}}^* - (Y_{\mathrm{c}}^* Y_{\mathrm{m}} - Y_{\mathrm{c}} Y_{\mathrm{m}}^*) \cos(2\varphi)]. \tag{56}
$$

The information of the system embedded in the reduced density matrices, Eqs. (43) and (50), can be used as a basic tool in estimating the entanglement between the subsystems. The

entanglement entropies take place in each subsystem as a signal of the entanglement between subsystems, provided that the associated reduced density matrix is non-zero. In the subsequent sections, these matrices will be used in dynamical analysis of entanglement on the basis of the entanglement entropies.

## 5 Linear entropy

Optical and mechanical subsystems share a non-separable quantum correlation through the entanglement between them, leading the system being distinctly nonclassical. An entangled quantum state cannot be represented as a simple product of the states of each subsystem. In such a case, the degree of entanglement can be estimated by, for example, entanglement entropies which are the most common entanglement measure [31]. It may be possible to quantify quantum entanglement contained in a state for many independently or identically distributed sub-quantum-systems, as well as the mutually coupled two subsystems. When estimating such entanglement for a subsystem, the information pertaining to the remaining part in the system is usually ignored.

The most basic entanglement entropy is the linear entropy, which is defined on the basis of the purity [18]. The linear entropy is zero for the case of pure states, whereas it is unity for completely mixed states. In general, we can represent the linear entropies of reduced states of which density matrices are $\rho_{n,l}^{\mathrm{R}}(X, X', t)$ as [28, 29]:

$$S_{L;n,l}(t) = 1 - \mathrm{Tr}[\rho_{n,l}^{\mathrm{R}}(t)]^2, \tag{57}$$

where

$$\mathrm{Tr}[\rho_{n,l}^{\mathrm{R}}(t)]^2 = \int_{-\infty}^{\infty} \int_{-\infty}^{\infty} \rho_{n,l}^{\mathrm{R}}(X, X', t) \rho_{n,l}^{\mathrm{R}}(X', X, t) dX' dX. \tag{58}$$

Let us first see the linear entropy for the mechanical subsystem in the ground state, whose density matrix is $\rho_{0,0}^{\mathrm{R}}(X_{\mathrm{m}}, X'_{\mathrm{m}}, t)$. By evaluating Eq. (58) using Eq. (43), we have the linear entropy for this case as

$$S_{L;0,0}(t) = 1 - \frac{N_{\mathrm{m}}}{\kappa_{\mathrm{m}}}, \tag{59}$$

where

$$\kappa_{\mathrm{m}} = \sqrt{4[\mathrm{Re}[A_{\mathrm{m}}]]^2 - B_{\mathrm{m}}^2}. \tag{60}$$

The linear entropy for the optical subsystem characterized by the reduced density matrix $\tilde{\rho}_{0,0}^{\mathrm{R}}(X_{\mathrm{c}}, X'_{\mathrm{c}}, t)$ can also be derived in a similar way using Eq. (50). This procedure gives

$$\tilde{S}_{L;0,0}(t) = 1 - \frac{N_{\mathrm{c}}}{\kappa_{\mathrm{c}}}, \tag{61}$$

where

$$\kappa_{\mathrm{c}} = \sqrt{4[\mathrm{Re}[A_{\mathrm{c}}]]^2 - B_{\mathrm{c}}^2}. \tag{62}$$

Thus, we confirm that the mathematical expressions of the two entropies, $S_{L;0,0}(t)$ and $\tilde{S}_{L;0,0}(t)$, are very similar to each other.

We can further investigate the linear entropy for diverse particular cases with a specific choice of time dependence for parameters, $\omega_{\mathrm{c}}(t)$, $\omega_{\mathrm{m}}(t)$, etc. Abundant physical phenomena associated with frequency modulations in optomechanical systems have been reported so far [32–37]. Quantum effects of optomechanical systems can be practically enhanced by periodic modulations of the frequencies [34–36]. For instance, arbitrary bosonic squeezing in coupled optomechanical systems can be achieved by modulating one or both frequencies

among the two which are associated with optical and mechanical modes respectively. Through this squeezing, it is possible to improve the measurement accuracy for weak signals [35, 36]. An optimal optomechanical-cooling scheme by suppressing the Stokes heating process via periodical modulations of the frequencies of cavity and mechanical resonators has also been proposed [37].

It is known that entanglement can also be improved by modulating optomechanical parameters, such as the frequencies [36], the coupling parameter [38–40] and the amplitude of the cavity mode laser [36, 41]. In order to see the influence of the periodical modulation of the optical frequency on the variation of the entanglement entropy, let us consider the case that $\omega_c(t)$ is modulated by a small sinusoidal perturbation, i.e., [28, 42]

$$\omega_c(t) = \omega_{c,0}[1 + \varepsilon \cos(\Omega t)], \tag{63}$$

where $\omega_{c,0}$, $\varepsilon$, and $\Omega$ are constants, while $\varepsilon \ll 1$. Meanwhile, we suppose, along with it, that the mechanical frequency does not depend on time:

$$\omega_m(t) = \omega_{m,0} \text{ (constant)}. \tag{64}$$

We can easily confirm that these suppositions make the system satisfy the adiabatic condition which was mentioned in Sec. 3 (see sentences given immediately after Eq. (19)). Then, from a minor evaluation, we have

$$\Delta(t) = \bar{\omega}_{c,0}[1 + \bar{\varepsilon} \cos(\Omega t)], \tag{65}$$

$$g(t) = g_0[1 + \varepsilon \cos(\Omega t)], \tag{66}$$

$$\varphi(t) \simeq \varphi_0 = \frac{1}{2}\text{atan}\left(\omega_{m,0}^{3/2}/\bar{\omega}_{c,0}^{1/2} - \bar{\omega}_{c,0}^{3/2}/\omega_{m,0}^{1/2}, 2g_0\right), \tag{67}$$

where $\bar{\omega}_{c,0} = \omega_{c,0} - \omega_L - \delta_{rp}$, $\bar{\varepsilon} = \varepsilon \omega_{c,0}/\bar{\omega}_{c,0}$, and $g_0 = \sqrt{\langle n_c \rangle \hbar/(m \omega_{m,0})} \omega_{c,0}/L$. Using Eqs. (65)-(67), Eqs. (18) and (19) can be rewritten as

$$\omega_{X,m} = \omega_{m,0} \cos^2 \varphi_0 + \frac{\bar{\omega}_{c,0}^2}{\omega_{m,0}}[1 + 2\bar{\varepsilon} \cos(\Omega t)] \sin^2 \varphi_0$$

$$+ g_0 \sqrt{\frac{\bar{\omega}_{c,0}}{\omega_{m,0}}}[1 + (\varepsilon + \bar{\varepsilon}/2) \cos(\Omega t)] \sin[2\varphi_0], \tag{68}$$

$$\omega_{X,c} = \frac{\omega_{m,0}^2}{\bar{\omega}_{c,0}}[1 - \bar{\varepsilon} \cos(\Omega t)] \sin^2 \varphi_0 + \bar{\omega}_{c,0}[1 + \bar{\varepsilon} \cos(\Omega t)] \cos^2 \varphi_0$$

$$- g_0 \sqrt{\frac{\omega_{m,0}}{\bar{\omega}_{c,0}}}[1 + (\varepsilon - \bar{\varepsilon}/2) \cos(\Omega t)] \sin[2\varphi_0]. \tag{69}$$

In the above equations, we have considered only up to the first order of $\varepsilon$ (and $\bar{\varepsilon}$) for simplicity.

Recall that the wave functions (and, consequently, the linear entropy) are represented in terms of the time functions, $\eta_m(t)$, $\eta_c(t)$, $\gamma_m(t)$, and $\gamma_c(t)$. Hence, in order to see the behavior of the linear entropy, it is necessary to derive the formula of them from the corresponding classical equations of motion. From the substitution of Eqs. (68) and (69) into Eqs. (A.9) and (A.10) in Appendix A, we have the formulae of $\omega_x$ and $\omega_q$. This leads the equations of motion given in Eqs. (A.11) and (A.12) in the form

$$\ddot{x} + [c_1 + c_2 \cos(\Omega t) + O(\varepsilon^2)]x = 0, \tag{70}$$

$$\ddot{q} + [\bar{c}_1 + \bar{c}_2 \cos(\Omega t) + O(\varepsilon^2)]q = 0, \tag{71}$$

where

$$c_1 = \omega_{m,0}^2 \cos^2 \varphi_0 + \bar{\omega}_{c,0}^2 \sin^2 \varphi_0 + g_0 \sqrt{\omega_{m,0}\bar{\omega}_{c,0}} \sin(2\varphi_0), \tag{72}$$

$$c_2 = 2\bar{\varepsilon}\bar{\omega}_{c,0}^2 \sin^2 \varphi_0 + g_0(\varepsilon + \bar{\varepsilon}/2)\sqrt{\omega_{m,0}\bar{\omega}_{c,0}} \sin(2\varphi_0), \tag{73}$$

$$\bar{c}_1 = \omega_{m,0}^2 \sin^2 \varphi_0 + \bar{\omega}_{c,0}^2 \cos^2 \varphi_0 - g_0 \sqrt{\omega_{m,0}\bar{\omega}_{c,0}} \sin(2\varphi_0), \tag{74}$$

$$\bar{c}_2 = 2\bar{\varepsilon}\bar{\omega}_{c,0}^2 \cos^2 \varphi_0 - g_0(\varepsilon + \bar{\varepsilon}/2)\sqrt{\omega_{m,0}\bar{\omega}_{c,0}} \sin(2\varphi_0). \tag{75}$$

The classical solutions of Eqs. (70) and (71) are given in terms of the Mathieu functions $Ce_\nu$ and $Se_\nu$, such that

$$x(t) = C_1 Ce_\nu\left(\frac{\Omega t}{2}, -\frac{2c_2}{\Omega^2}\right) + C_2 Se_\nu\left(\frac{\Omega t}{2}, -\frac{2c_2}{\Omega^2}\right), \tag{76}$$

$$q(t) = \bar{C}_1 Ce_{\bar{\nu}}\left(\frac{\Omega t}{2}, -\frac{2\bar{c}_2}{\Omega^2}\right) + \bar{C}_2 Se_{\bar{\nu}}\left(\frac{\Omega t}{2}, -\frac{2\bar{c}_2}{\Omega^2}\right), \tag{77}$$

where $\nu = 4c_1/\Omega^2$ and $\bar{\nu} = 4\bar{c}_1/\Omega^2$, whereas $C_i$ ($i = 1, 2$) and $\bar{C}_i$ are constants. For the characteristics of the Mathieu functions including their stability, refer, for example, to Refs. [42, 43].

We can also represent the solutions in another form as given in Eqs. (A.13) and (A.14) in Appendix A (e.g., see Ref. [42]), with

$$\eta_m(t) = \left[Ce_\nu^2\left(\frac{\Omega t}{2}, -\frac{2c_2}{\Omega^2}\right) + Se_\nu^2\left(\frac{\Omega t}{2}, -\frac{2c_2}{\Omega^2}\right)\right]^{1/2}, \tag{78}$$

$$\eta_c(t) = \left[Ce_{\bar{\nu}}^2\left(\frac{\Omega t}{2}, -\frac{2\bar{c}_2}{\Omega^2}\right) + Se_{\bar{\nu}}^2\left(\frac{\Omega t}{2}, -\frac{2\bar{c}_2}{\Omega^2}\right)\right]^{1/2}, \tag{79}$$

$$\gamma_m(t) = atan\left(Ce_\nu\left(\frac{\Omega t}{2}, -\frac{2c_2}{\Omega^2}\right), Se_\nu\left(\frac{\Omega t}{2}, -\frac{2c_2}{\Omega^2}\right)\right), \tag{80}$$

$$\gamma_c(t) = atan\left(Ce_{\bar{\nu}}\left(\frac{\Omega t}{2}, -\frac{2\bar{c}_2}{\Omega^2}\right), Se_{\bar{\nu}}\left(\frac{\Omega t}{2}, -\frac{2\bar{c}_2}{\Omega^2}\right)\right). \tag{81}$$

We have depicted the time behavior of the linear entropies for this case in Fig. 2 using the formulae of time functions given in Eqs. (78)-(81). From a close inspection of Fig. 2, we confirm that the linear entropies for the mechanical and the optical parts coincide each other. Evidently, the entanglement is shared between the two subsystems through their coupling.

Figure 2 shows additional diverse properties of the linear entropy depending on several different values of parameters. We can see from Fig. 2(A) that the linear entropies emerge from zero as the coupling strength $g_0$ grows; in addition, their mean values increase according to the growth of $g_0$. From this, we can conclude that the entanglement is relatively high when the coupling between the two subsystems is strong. This consequence agrees with the result of Ref. [14] in which entanglement was analyzed using other means. Besides, the linear entropies fluctuate in time in a periodic fashion. Such fluctuations also become significant as $g_0$ grows.

The effects of $\varepsilon$ on the linear entropies can be seen from Fig. 2(B). While the linear entropies are independent of time when $\varepsilon = 0$, they fluctuate unless $\varepsilon = 0$ and such fluctuations gradually augment as $\varepsilon$ increases. On the other hand, Fig. 2(C) shows that the fluctuations of the linear entropies do not monotonically increase as $\Omega$ grows. The fluctuations of the linear entropy become greater at first in response to the grow of $\Omega$ from 1.5, but they eventually collapse for a higher value of $\Omega$ (see the green curve in Fig. 2(C)).

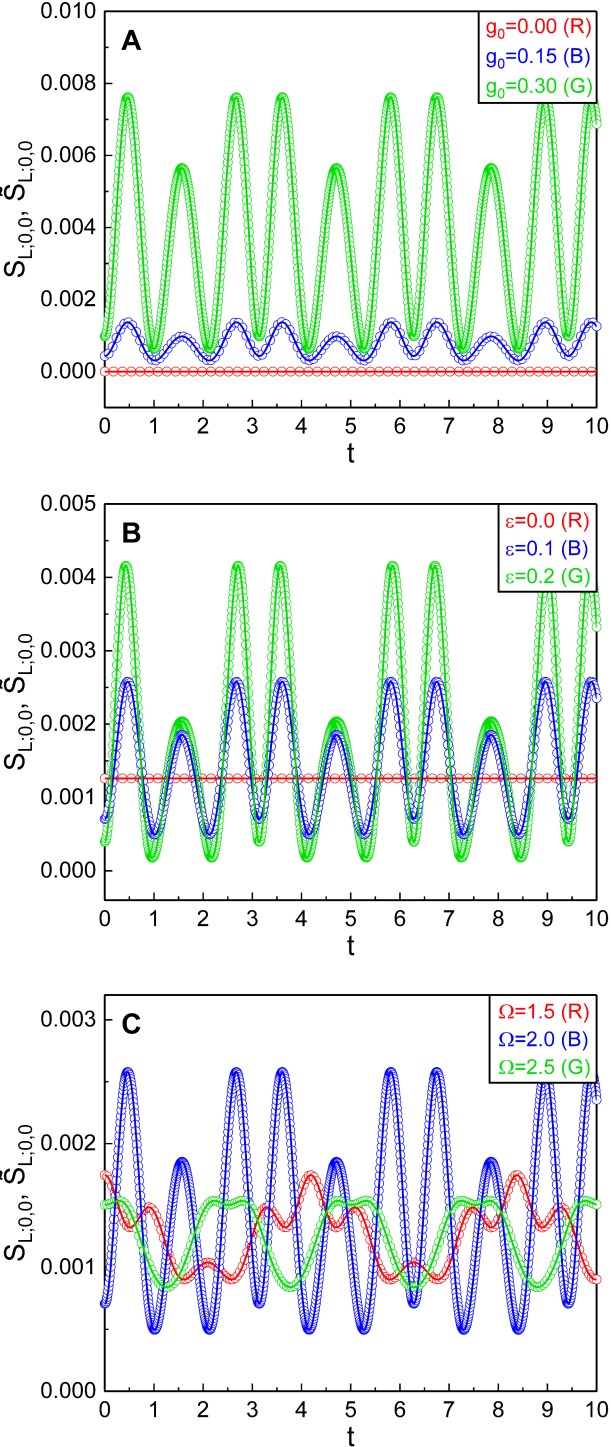

Figure 2: Temporal evolution of linear entropies $S_{L;0,0}(t)$ (solid lines) and $\tilde{S}_{L;0,0}(t)$ (circles) for several different values of $g_0$ (A), $\varepsilon$ (B), and $\Omega$ (C). The legend of (A) means that $g_0 = 0.00$ for red curves, $g_0 = 0.15$ for blue curves, and $g_0 = 0.30$ for green curves. The legends of other panels are also interpreted in this way. We used $(\varepsilon, \Omega)=(0.1, 2)$ for (A), $(g_0, \Omega)=(0.2, 2)$ for (B), and $(g_0, \varepsilon)=(0.2, 0.1)$ for (C). All other values are common and given by $\omega_{m,0} = 1$, $\bar{\omega}_{c,0} = 3$, and $\nu \equiv \bar{\varepsilon}/\varepsilon = 1.2$.

# 6 Von Neumann entropy and Rényi entropy

As well as the linear entropy, the von Neumann entropy is also an essential tool for quantifying quantum information contained in states [19]. The von Neumann entropy is a general entanglement entropy and it is known as the quantum counterpart of the classical Shannon entropy [44]. The von Neumann entropies for the mechanical and the optical parts are defined, respectively, as [20,30]

$$S_{N;n,l} = -\text{Tr}[\rho_{n,l}^{\text{R}}(X_{\text{m}}, X_{\text{m}}', t) \ln \rho_{n,l}^{\text{R}}(X_{\text{m}}, X_{\text{m}}', t)], \tag{82}$$

$$\tilde{S}_{N;n,l} = -\text{Tr}[\tilde{\rho}_{n,l}^{\text{R}}(X_{\text{c}}, X_{\text{c}}', t) \ln \tilde{\rho}_{n,l}^{\text{R}}(X_{\text{c}}, X_{\text{c}}', t)]. \tag{83}$$

Another useful class of entanglement entropy is the Rényi entropy. This entropy is a generalization of the von Neumann entropy [19, 30, 45]: in fact, the Rényi entropy is the most general type of entropy that is used for measuring entanglement. The Reńyi entropies of order $\alpha$ (Reńyi-$\alpha$ entropies), respectively for the mechanical and the optical parts, are given by

$$S_{\alpha;n,l} = \frac{1}{1-\alpha} \ln[\text{Tr}[\rho_{n,l}^{\text{R}}(X_{\text{m}}, X_{\text{m}}', t)]^{\alpha}], \tag{84}$$

$$\tilde{S}_{\alpha;n,l} = \frac{1}{1-\alpha} \ln[\text{Tr}[\tilde{\rho}_{n,l}^{\text{R}}(X_{\text{c}}, X_{\text{c}}', t)]^{\alpha}], \tag{85}$$

where $\alpha > 0$ and $\alpha \neq 1$. Equations (84) and (85) are defined in a way that the Rényi entropies preserve the additivity for independent events according to the axiom of probability. In particular, Rényi entropy of order 2 provides a useful measure of quantum information for multimode Gaussian states, which can be adopted as a privileged tool for addressing related correlations via entanglement [19].

We also focus on the ground state for both von Neumann and Reńyi entropies. We consider the spectral decompositions [20,30,46] of the reduced density matrices of the subsystems as a tackle for obtaining these entropies. Such decompositions for mechanical and optical parts can be carried out starting from the eigenvalue equations of the form, respectively

$$\int_{-\infty}^{\infty} dX_{\text{m}}' \rho_{0,0}^{\text{R}}(X_{\text{m}}, X_{\text{m}}', t) f_j(X_{\text{m}}', t) = p_j(t) f_j(X_{\text{m}}, t), \tag{86}$$

$$\int_{-\infty}^{\infty} dX_{\text{c}}' \tilde{\rho}_{0,0}^{\text{R}}(X_{\text{c}}, X_{\text{c}}', t) \tilde{f}_k(X_{\text{c}}', t) = \tilde{p}_k(t) \tilde{f}_k(X_{\text{c}}, t). \tag{87}$$

From a straightforward evaluation for these equations, we have

$$p_j(t) = [1 - \xi_{\text{m}}(t)]\xi_{\text{m}}^j(t), \tag{88}$$

$$\tilde{p}_k(t) = [1 - \xi_{\text{c}}(t)]\xi_{\text{c}}^k(t), \tag{89}$$

$$f_j(X_{\text{m}}, t) = \sqrt[4]{\frac{\kappa_{\text{m}}}{\pi}} \frac{1}{\sqrt{2^j j!}} H_j[\sqrt{\kappa_{\text{m}}} X_{\text{m}}] \exp\{-[\kappa_{\text{m}}/2 + i\text{Im}[A_{\text{m}}]]X_{\text{m}}^2\}, \tag{90}$$

$$\tilde{f}_k(X_{\text{c}}, t) = \sqrt[4]{\frac{\kappa_{\text{c}}}{\pi}} \frac{1}{\sqrt{2^k k!}} H_k[\sqrt{\kappa_{\text{c}}} X_{\text{c}}] \exp\{-[\kappa_{\text{c}}/2 + i\text{Im}[A_{\text{c}}]]X_{\text{c}}^2\}, \tag{91}$$

where

$$\xi_{\text{m}}(t) = \frac{B_{\text{m}}}{2\text{Re}[A_{\text{m}}] + \kappa_{\text{m}}}, \tag{92}$$

$$\xi_{\text{c}}(t) = \frac{B_{\text{c}}}{2\text{Re}[A_{\text{c}}] + \kappa_{\text{c}}}. \tag{93}$$

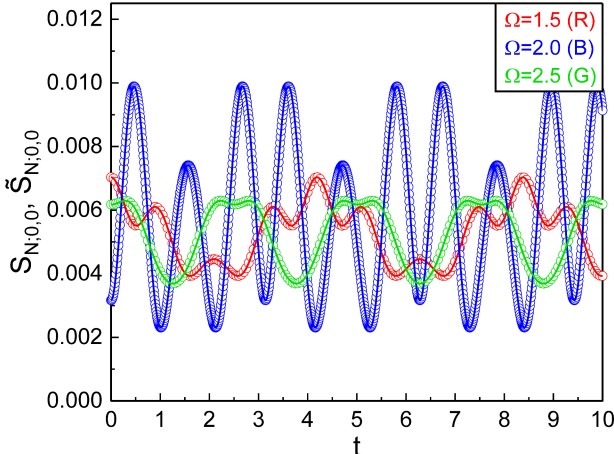

Figure 3: Temporal evolution of von Neumann entropies $S_{N;0,0}(t)$ (solid lines) and $\tilde{S}_{N;0,0}(t)$ (circles) for several different values of $\Omega$ as designated in the legend. We used $g_0 = 0.2$, $\varepsilon = 0.1$, $\omega_{m,0} = 1$, $\bar{\omega}_{c,0} = 3$, and $\nu \equiv \bar{\varepsilon}/\varepsilon = 1.2$.

Then, as is well known [30], it is possible to carry out the spectral decompositions of the reduced density matrices, leading to

$$\rho_{0,0}^{R}(X_m, X_m', t) = \sum_{j=0}^{\infty} p_j(t) f_j^*(X_m', t) f_j(X_m, t), \tag{94}$$

$$\tilde{\rho}_{0,0}^{R}(X_c, X_c', t) = \sum_{k=0}^{\infty} \tilde{p}_k(t) \tilde{f}_k^*(X_c', t) \tilde{f}_k(X_c, t). \tag{95}$$

Using Eqs. (94) and (95), we readily have the formulae of the von Neumann and Rényi entropies, such that

$$S_{N;0,0} = -\ln(1 - \xi_m) - \frac{\xi_m}{1 - \xi_m} \ln \xi_m, \tag{96}$$

$$\tilde{S}_{N;0,0} = -\ln(1 - \xi_c) - \frac{\xi_c}{1 - \xi_c} \ln \xi_c, \tag{97}$$

$$S_{\alpha;0,0} = \frac{1}{1 - \alpha} \ln \frac{(1 - \xi_m)^{\alpha}}{1 - \xi_m^{\alpha}}, \tag{98}$$

$$\tilde{S}_{\alpha;0,0} = \frac{1}{1 - \alpha} \ln \frac{(1 - \xi_c)^{\alpha}}{1 - \xi_c^{\alpha}}. \tag{99}$$

If the time dependence of $\omega_m(t)$, $\Delta(t)$, and $g(t)$ disappears, the outcome, Eqs. (96) and (97), reduces to that in Refs. [20, 46]. For an other simple case, the results, Eqs. (96)-(99), are similar to those of Ref. [30] but not exactly the same.

The time behavior of the von Neumann entropies is shown in Fig. 3 for several different values of $\Omega$. As you can see, the pattern of this behavior is very much the same as that of Fig. 2(C) which corresponds to the linear entropies. In fact, the linear entropy is an approximation of the von Neumann entropy [18]. The value of the linear entropy is restricted within the range from 0 to 1. However, the upper bound of the von Neumann entropy is not so simply determined (see, e.g., Ref. [47]).

Temporal evolution of the Rényi entropies for several different values of $\alpha$ are given in Fig. 4. We see that Rényi entropies become small as $\alpha$ increases. However, such entropy changes

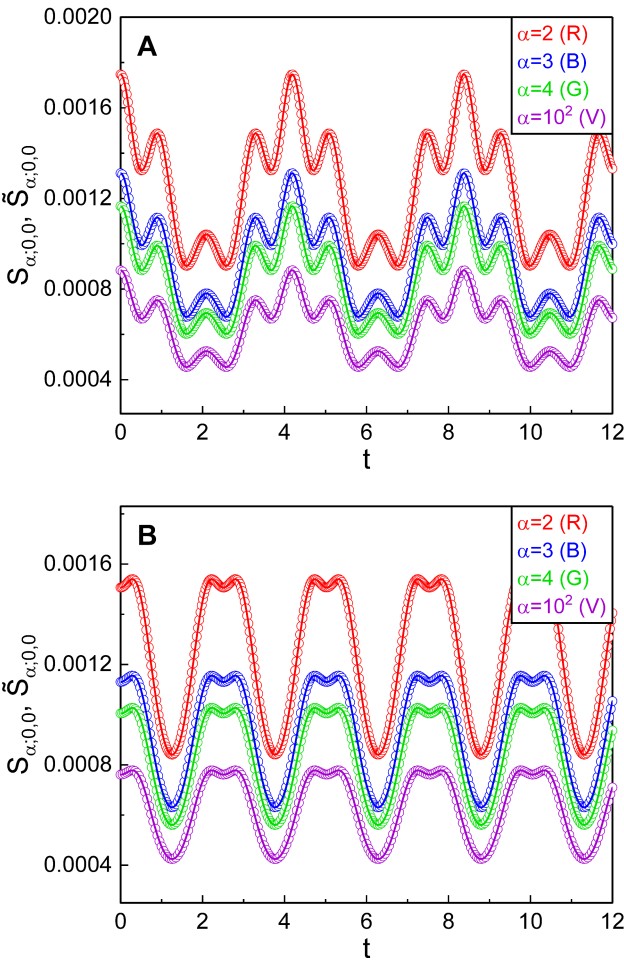

Figure 4: Temporal evolution of Rényi entropies $S_{\alpha;0,0}(t)$ (solid lines) and $\tilde{S}_{\alpha;0,0}(t)$ (circles) for several different values of $\alpha$. We used $\Omega = 1.5$ for (A) and $\Omega = 2.5$ for (B). All other values are common and they are $g_0 = 0.2$, $\varepsilon = 0.1$, $\omega_{m,0} = 1$, $\bar{\omega}_{c,0} = 3$, and $\nu \equiv \bar{\varepsilon}/\varepsilon = 1.2$.

per unit increase of $\alpha$ is not so large when the value of $\alpha$ is sufficiently high. Consequently, the Rényi entropies reduce to their minimum values in the limit $\alpha \to \infty$. The Rényi entropies also exhibit a periodical variation in time. The patterns of such a variation shown in Figs. 4(A) and 4(B) quite resemble those of Fig. 3 (or Fig. 2(C)) with $\Omega = 1.5$ and $\Omega = 2.5$, respectively.

We see from Fig. 5 that each kind of entanglement entropies grows as the coupling strength $g_0$ increases. The von Neumann entropy among the three kinds of entropies exhibits the highest rate of increase in such a growth. The rate of growth of the Rényi entropy with $\alpha = 2$ is almost the same as that of the linear entropy. However, strictly speaking, the growth of the Rényi-2 entropy is slightly faster than that of the linear entropy. Rényi-2 entropy is an alternative form of the linear entropy, whereas Rényi-1/2 entropy implies quantum uncertainty defined in terms of the skew information [48].

Although we have evaluated entanglement entropies for the case of the ground state of the optical (and mechanical) oscillators partly for convenience, it may highly be possible to think of an excited state of the optical oscillator, because it is driven by a laser field. If such a state is far from the ground state, the entanglement between the optical and the mechanical modes may be enhanced due the increase of the quadrature uncertainty in the optical mode. Notice that, if the quantum number in a coupled oscillatory motion is large, the entanglement

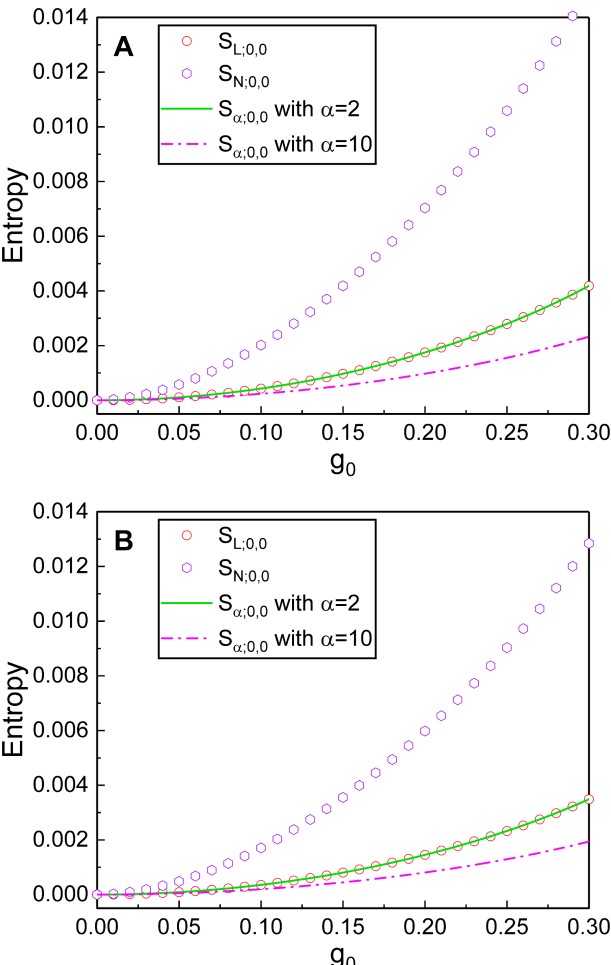

Figure 5: The increment of entanglement entropies $S_{L;0,0}(t)$, $S_{N;0,0}(t)$, and $S_{\alpha;0,0}(t)$ at $t = 0$ for (A) and at $t = 1$ for (B) through the increase of coupling strength $g_0$. We used $\varepsilon = 0.1$, $\Omega = 1.5$, $\omega_{m,0} = 1$, $\bar{\omega}_{c,0} = 3$, and $\nu \equiv \bar{\varepsilon}/\varepsilon = 1.2$.

between the associated subsystems is enhanced [49–51].

# 7  Conclusion

We have investigated entanglement entropies for a nanomechanical resonator coupled to a cavity field. The effects of the time variation of parameters, such as cavity frequency $\omega_c(t)$ and the coupling strength $g(t)$, on the evolution of the entanglement entropies have been analyzed in detail. The wave functions in Fock state and the corresponding reduced density matrices of the coupled system have been evaluated under the assumption that the system evolves adiabatically. Using the reduced density matrices, we have derived analytical formulae of several fundamental entanglement entropies, such as linear entropies, von Neumann entropies, and Rényi entropies. These entropies emerge and increase as the coupling strength $g_0$ becomes high, while they disappear for uncoupled systems.

In particular, we have focused our research on the effects of a modulation in the cavity frequency: The cavity frequency is perturbed by a sinusoidally varying small modulation term of which frequency is $\Omega$. If $g_0$ and/or the amplitude $\varepsilon$ of the modulation term in $\omega_c(t)$ grow, the fluctuations of the entropies also increase. However, such fluctuations do not monotonically

increase along the grow of the frequency $\Omega$. Although fluctuations of the entropies increase as $\Omega$ grows provided that $\Omega$ is sufficiently low, the fluctuations rather reduce as $\Omega$ reaches a higher value.

From the graphical analyses for the time evolution of the entropies, we have confirmed that the entanglement entropies of the mechanical part are the same as those of the optical part. This consequence stems from the fact that the entanglement entropies are state quantities shared between the nanoresonator and the cavity field. The fluctuation patterns of the three kinds of entropies resemble one another. However, the entanglement entropies grow as $g_0$ increases with different rates depending on the type of entropies; we confirmed that the von Neumann entropy exhibits the highest rate of increase.

Our technique for characterizing entanglement entropies can also be applied to other states beyond the Fock states, such as the coherent states, squeezed states, and thermal states. As well as theoretical quantification of entanglement, experimental measurement of entanglement or entanglement entropies may also be a major concern in this context. Recently, protocols for measuring entanglement entropies through an optimized universal tool have been proposed [31,45].

As a final remark, we have rigorously analyzed entanglement entropies which are necessary for the understanding of entanglement dynamics related to quantum optical control of optomechanical nano-devices. We hope that this work can further motivate great ideas in achieving strong entanglement based on optomechanical coupling, which plays a key role in the field of quantum information, such as quantum computation protocols [52], quantum secure communication [53], teleportation [54,55], and neo-cryptographic systems [56].

## Acknowledgements

This work was supported by the National Research Foundation of Korea(NRF) grant funded by the Korea government(MSIT) (No.: NRF-2021R1F1A1062849).

## A Derivation of wave functions in the transformed system

Let us consider a time-dependent harmonic oscillator of which Hamiltonian is given in terms of a coordinate operator $\hat{y}$ by

$$\hat{H} = \frac{1}{2}[A(t)\hat{y}^2 + B(t)\hat{p}_y^2], \tag{A.1}$$

where $\hat{p}_y = -i\hbar \partial/\partial y$, while $A(t)$ and $B(t)$ are time functions that are differentiable with respect to time. The corresponding classical equation of motion can be written as

$$\ddot{y} - \frac{\dot{B}(t)}{B(t)}\dot{y} + A(t)B(t)y = 0. \tag{A.2}$$

Without loss of generality, let us express the classical solution of the above equation in the form

$$y(t) = \eta(t)[C_+ e^{i\gamma(t)} + C_- e^{-i\gamma(t)}], \tag{A.3}$$

where $\eta(t)$ and $\gamma(t)$ are real time functions and $C_\pm$ are real constants. Then, the quantum wave functions in the Fock state are given by [57,58]

$$\Psi_n(y,t) = \Phi_n(y,t)\exp[i\delta_n(t)], \tag{A.4}$$

where $\Phi_n(y, t)$ are the eigenstates which are given in terms of $\eta(t)$ and $\gamma(t)$, such that

$$\Phi_n(y, t) = \sqrt[4]{\frac{\dot\gamma(t)}{\hbar\pi B(t)}}\frac{1}{\sqrt{2^n n!}}H_n\left(\sqrt{\frac{\dot\gamma(t)}{\hbar B(t)}}y\right)\exp\left[-\frac{1}{2\hbar B(t)}\left(\dot\gamma(t) - i\frac{\dot\eta(t)}{\eta(t)}\right)y^2\right], \quad (A.5)$$

while the phases are given by $\delta_n(t) = -(n + 1/2)\gamma(t)$.

Now, let us turn our attention to the optomechanical system given in the text. For the mechanical part of the system, the dimensionless position operator of the nanomechanical resonator and its canonical conjugate operator are given by

$$\hat{X}_{\mathrm{m}} = \sqrt{\frac{m\omega_{\mathrm{m}}(t)}{\hbar}}\hat{x} \qquad \hat{P}_{\mathrm{m}} = \sqrt{\frac{1}{m\omega_{\mathrm{m}}(t)\hbar}}\hat{p}_x, \quad (A.6)$$

where $\hat{x}$ and $\hat{p}_x(= -i\hbar\partial/\partial x)$ are ordinary position and momentum operators of the resonator. Similarly, the operator of the dimensionless quadrature and its canonical conjugate variable in the optical part are represented as

$$\hat{X}_{\mathrm{c}} = \sqrt{\frac{\epsilon_0\Delta(t)}{\hbar}}\hat{q} \qquad \hat{P}_{\mathrm{c}} = \sqrt{\frac{1}{\epsilon_0\Delta(t)\hbar}}\hat{p}_q, \quad (A.7)$$

where $\hat{q}$ and $\hat{p}_q(= -i\hbar\partial/\partial q)$ are the ordinary quadrature operator and its canonical conjugate variable, respectively. Using these, the Hamiltonian in Eq. (20) can be rewritten in terms of ordinary coordinates in the form

$$\hat{\mathcal{H}} = \frac{1}{2}\left(m\omega_x^2(t)\hat{x}^2 + \frac{\hat{p}_x^2}{m} + \epsilon_0\omega_q^2(t)\hat{q}^2 + \frac{\hat{p}_q^2}{\epsilon_0}\right), \quad (A.8)$$

where

$$\omega_x(t) = \sqrt{\omega_{X,\mathrm{m}}(t)\omega_{\mathrm{m}}(t)}, \quad (A.9)$$

$$\omega_q(t) = \sqrt{\omega_{X,\mathrm{c}}(t)\Delta(t)}. \quad (A.10)$$

The corresponding equations of motion are given by

$$\ddot{x} + \omega_x^2(t)x = 0, \quad (A.11)$$

$$\ddot{q} + \omega_q^2(t)q = 0. \quad (A.12)$$

The classical solutions of these two equations are of the form

$$x(t) = \eta_{\mathrm{m}}(t)[C_+ e^{i\gamma_{\mathrm{m}}(t)} + C_- e^{-i\gamma_{\mathrm{m}}(t)}], \quad (A.13)$$

$$q(t) = \eta_{\mathrm{c}}(t)[\bar{C}_+ e^{i\gamma_{\mathrm{c}}(t)} + \bar{C}_- e^{-i\gamma_{\mathrm{c}}(t)}]. \quad (A.14)$$

The wave functions for the transformed Hamiltonian can be divided into $x$ and $q$ parts; the $x$ part is given in terms of $\eta_{\mathrm{m}}(t)$ and $\gamma_{\mathrm{m}}(t)$ whereas the $q$ part is in terms of $\eta_{\mathrm{c}}(t)$ and $\gamma_{\mathrm{c}}(t)$. For the $x$ part, the use of the relation $B = 1/m$ leads to

$$\Phi_n(x, t) = \sqrt[4]{\frac{m\dot\gamma_{\mathrm{m}}(t)}{\hbar\pi}}\frac{1}{\sqrt{2^n n!}}H_n\left(\sqrt{\frac{m\dot\gamma_{\mathrm{m}}(t)}{\hbar}}x\right)\exp\left[-\frac{m}{2\hbar}\left(\dot\gamma_{\mathrm{m}}(t) - i\frac{\dot\eta_{\mathrm{m}}(t)}{\eta_{\mathrm{m}}(t)}\right)x^2\right], \quad (A.15)$$

while, from $B = 1/\epsilon_0$ for the $q$ part, we have

$$\tilde{\Phi}_l(q, t) = \sqrt[4]{\frac{\epsilon_0\dot\gamma_{\mathrm{c}}(t)}{\hbar\pi}}\frac{1}{\sqrt{2^l l!}}H_l\left(\sqrt{\frac{\epsilon_0\dot\gamma_{\mathrm{c}}(t)}{\hbar}}q\right)\exp\left[-\frac{\epsilon_0}{2\hbar}\left(\dot\gamma_{\mathrm{c}}(t) - i\frac{\dot\eta_{\mathrm{c}}(t)}{\eta_{\mathrm{c}}(t)}\right)q^2\right]. \quad (A.16)$$

If we represent the eigenstates, Eqs. (A.15) and (A.16), in terms of dimensionless canonical variables using Eqs. (A.6) and (A.7), we have Eqs. (25) and (27) in the text. Notice that the eigenstates described in terms of the dimensionless canonical variable $X_m$ (Eq. (25)) are dimensionless, while the dimension of Eq. (A.15) is $L^{-1/2}$ where L is the length dimension. According to this, we have adjusted the normalization factor of Eq. (25) so that it becomes dimensionless. Similar adjustment has also been applied in Eq. (27) which corresponds to the optical part.

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
