# Peer review of "Entropic analysis of optomechanical entanglement for a nanomechanical resonator coupled to an optical cavity field"

_SciPost Physics Core, doi:SciPost Phys. Core 4, 024 (2021)_

## Round 1 · Referee Report · Anonymous (Referee 1) · 2020-11-25

Weaknesses

1) The analysis is very similar to what has been presented in previous works.

2) No real physical motivation for the studied problem of a modulated optical cavity frequency.

3) Not applicable to realistic optomechanical systems with dissipation.

Report

In this paper different entanglement measures for a linearized optomechanical system are evaluated. Analytic expressions for the linear entropy, the von Neumann entropy and the Renyi entropy are derived from the exact evolution of the wavefunction of the combined system. The resulting entropies are then investigated for a system with a modulated frequency and for different coupling strengths.

The generation of entangled states in optomechanical systems is a well-studied problem, which in the considered linearized regime reduces to the study of the entanglement of a two-mode Gaussian state. Although in the quantum optics literature one usually doesn't evaluate the von Neumann entropy or the Renyi entropy explicitly, also the general expressions for these entropies in a two-mode system are known. Indeed, the current analysis very similar to what is already presented, for example, in Ref. [30]. Since only the coherent evolution is considered the whole problem discussed in this paper reduces the evaluation of the wavefunction of two coupled oscillators. Although the analytic results for the different entropies etc. might still be involved, their derivation is conceptually straightforward.

Despite the claim of providing more insights into entanglement properties of optomechanical systems, the actual findings have little relevance for such systems. The important influence of dissipation or thermal noise, but also the preparation of the initial state or the readout of the entanglement are not discussed at all. The study of a modulated cavity frequency is not motivated and no connection to any quantum information processing applications is made.

In summary, I fail to see a significant new theoretical result or an important physical insight for the field of optomechanics. Therefore, I cannot recommend a publication of this work in SciPost.

  • validity: high
  • significance: poor
  • originality: low
  • clarity: ok
  • formatting: good
  • grammar: excellent

Author:  Jeong Ryeol Choi  on 2020-12-27  [id 1112]

(in reply to Report 1 on 2020-11-25)

<1. Response to the comment "Indeed, the current analysis very similar to what is already presented, for example, in Ref. [30].">
The mathematical evaluations given in Ref. [30] is based on the simple rotation method. I have pointed out the weak point of the simple rotation method in my manuscript (see from line 8 on page 4 to the last line on the same page). Usually, the simple rotation method is inapplicable unless there are some restriction(s) in the coupled oscillatory systems.
The system adopted in Ref. [30] is a simple case where the masses of the two oscillatory subsystems are identical to each other: m_1 = m_2 = 1. In this restricted case, the simple rotation method is applicable. However, for the case that m_1 and m_2 are not equal to each other, such a simple rotation method is no longer applicable (see the appended file for detailed verification of this). In addition to this, the simple rotation method is also not applicable to the system in the present work, as mentioned in my manuscript with a verification.
In order to overcome this difficulty, I have adopted much more powerful method which is the unitary transformation method in my manuscript instead of the simple rotation method.

<2. Response to the comment "Not applicable to realistic optomechanical systems with dissipation.">
In the revised version, I will improve the manuscript in such a way that it can also be applicable to dissipative optomechanical systems.

<3. Response to the comment "No real physical motivation for the studied problem of a modulated optical cavity frequency.">
In the revised version, I will provide physical motivation for the modulation of the optical cavity frequency.

Attachment:

ede.pdf

---

## Round 1 · Referee Report · Anonymous (Referee 2) · 2020-12-8

Report

The author calculates analytically the linear, von Neumann and Rényi entanglement entropies for a system composed by two coupled harmonic oscillators with time dependent parameters in the adiabatic regime. The harmonic oscillators describe a mechanical resonator and a optical cavity as in Ref.[2]. The entanglement entropies increase as the coupling strength increases, as expected.
The calculations are sound and elegant, I recommend publication in SciPost Physics after the requested changes are addressed.

Requested changes

1) validity of the adiabatic approximation: the author should quantify what "the variation of φ(t) over time is sufficiently slow" (as written after eq. (19)) means. In particular the author should justify analytically or numerically in which parameters regime the adiabatic approximation is valid, and check that the parameters used in Sec. 5 and Sec. 6 fulfill it.

2) the derivative of β(t) does not appear in eq.(17), is this an exact result or it depends on some approximation? if it depends on some approximation it should be pointed out in the text.

3) a more self-contained description of the system in Sec.2 may be useful to the reader. At the moment the main reference for the description of the system is Ref.[2].

4) the entanglement entropies are calculated for the ground states. It is not clear if both systems can actually be in the ground state since the optical cavity is driven by a laser. The author should justify that.

5) Can the author give an idea of what would change for excited states?

  • validity: good
  • significance: good
  • originality: good
  • clarity: good
  • formatting: excellent
  • grammar: good

Author:  Jeong Ryeol Choi  on 2020-12-27  [id 1113]

(in reply to Report 2 on 2020-12-08)

<Response to the comment of the reviewer 2>
In the revised version, I will improve the manuscript regarding the report of the reviewer 2.

---

## Round 2 · Referee Report · Anonymous (Referee 2) · 2021-5-14

Report

While the Author answered reasonably to my points of the first review, I find that his description of dissipative effects is quite lacking.
If I follow correctly what the Author does: the dumping therms are "phenomenologically" added to the Hamiltonian, and then approximated away obtaining the same result of the previous version. So, nothing changes except some new "phenomenological terms" are added and removed without the possibility to even understand if they were correct. If this is the case, I think that this must be improved before pubblication.

Requested changes

1- Improve or remove the description of the description of the dissipative effects.

  • validity: good
  • significance: good
  • originality: good
  • clarity: good
  • formatting: excellent
  • grammar: excellent

Author:  Jeong Ryeol Choi  on 2021-05-21  [id 1444]

(in reply to Report 1 on 2021-05-14)

I will revise the manuscript according to the reviewer's comment.

---

## Round 2 · Author Response

Dear editor,
I am re-submitting a paper entitled "Entropic analysis of optomechanical entanglement for a nanomechanical resonator coupled to an optical cavity field" in SciPost Physics Core.

I agree with this submission.
This work is original research and has not been published or submitted for publication elsewhere.
I declare no conflict of interests.

Author: Jeong Ryeol Choi
Author’s contact information:
Affiliation: Department of Nanoengineering, Kyonggi University, Yeongtong-gu, Suwon, Kyeonggi-do, 16227, Republic of Korea
E-mail address: choiardor@hanmail.net
Tel: +82 31 249 1320
Fax: +82 31 249 9604

Jeong Ryeol Choi
Department of Nanoengineering, Kyonggi University
Republic of Korea

---

## Round 2 · List of Changes

<List of Correction>

  1. Line 12 on page 3. [Old] The relation between the optical frequency \Delta and the cavity frequency \omega_c is given by \Delta = \omega_c - \omega_L - \delta_{rp}, where \delta_{rp} is the shift of the cavity frequency by radiation pressure. On the other hand, the coupling strength is given by g(t) = G(t) \sqrt{<n_c>}, where G(t) = [\omega_c(t)/L(t)]\sqrt{\hbar/[m\omega_m(t)]}, m is effective mass of the resonator, L is the cavity length, and <n_c> is the mean cavity photon number. For a more detailed description of the system, refer to Ref. [2]. [New] If we consider that cavity is driven by a laser field, the relation between the optical frequency \Delta and the cavity frequency \omega_c is given by \Delta = \omega_c - \omega_L - \delta_{rp}, where \delta_{rp} is the shift of the cavity frequency by radiation pressure [2]. On the other hand, the coupling strength is given by g(t) = G(t) \sqrt{<n_c>}, where G(t) = [\omega_c(t)/L(t)]\sqrt{\hbar/[m\omega_m(t)]}, m is effective mass of the resonator, L is the cavity length, and <n_c> is the mean cavity photon number [2].

  2. Equation 3 and subsequent equations are revised by introducing damping constants \zeta_m and \zeta_c so that they can also be applied to dissipative optomechanical systems.

  3. Last line of Eq. (17) and a subsequent sentence on page 5. [Old] -\hbar\dot{\varphi}(t)[\beta(t)P_mX_c-\beta^{-1}(t)P_cX_m], where [New] -\hbar[\dot{\varphi}_1(t)P_mX_c-\dot{\varphi}_2(t)P_cX_m], where \varphi_1(t) = \varphi(t)\beta(t), \varphi_2(t)=\varphi(t)\beta^{-1}(t), and

  4. After Eq. (19) on page 5. [Old] Let us assume that the variation of \varphi(t) over time is sufficiently slow. [New] Let us assume that the variations of \varphi_1(t) and \varphi_2(t) over time are sufficiently slow. This means weak damping, \zeta_m(t) ~ 0 and \zeta_c(t) ~ 0, in addition to the previous assumption that the variations of g(t), \Delta(t), and \omega_m(t) are slow.

  5. Line 3 from bottom on page 9. [Old] We can further investigate the linear entropy for diverse particular cases with a specific choice of time dependence for parameters, ω_c (t), ω_m (t), etc. For instance, let us consider … [New] We can further investigate the linear entropy for diverse particular cases with a specific choice of time dependence for parameters, ω_c (t), ω_m (t), etc. Abundant physical phenomena associated with frequency modulations in optomechanical systems have been reported so far [32-37]. Quantum effects of optomechanical systems can be practically enhanced by periodic modulations of the frequencies [34-36]. For instance, arbitrary bosonic squeezing in coupled optomechanical systems can be achieved by modulating one or both frequencies among the two which are associated with optical and mechanical modes respectively. Through this squeezing, it is possible to improve the measurement accuracy for weak signals [35,36]. An optimal optomechanical-cooling scheme by suppressing the Stokes heating process via periodical modulations of the frequencies of cavity and mechanical resonators has also been proposed [37]. It is known that entanglement can also be improved by modulating optomechanical parameters, such as the frequencies [36], the coupling parameter [38-40] and the amplitude of the cavity mode laser [36,41]. In order to see the influence of the periodical modulation of the optical frequency on the variation of the entanglement entropy, let us consider …

  6. After Eq. (64) on page 10. [Old] Then, from a minor evaluation, … [New] We can easily confirm that these suppositions make the system satisfy the adiabatic condition which was mentioned in Sec. 3 (see sentences given immediately after Eq. (19)). Then, from a minor evaluation, …

  7. Line 3 on page 17. [Old] No sentences. [New] Although we have evaluated entanglement entropies for the case of the ground state of the optical (and mechanical) oscillators for convenience in part, it may highly be possible to think of an excited state of the optical oscillator, because it is driven by a laser field. If such a state is far from the ground state, the entanglement between the optical and the mechanical modes may be enhanced due the increase of the quadrature uncertainty in the optical mode. Notice that, if the quantum number in a coupled oscillatory motion is large, the entanglement between the associated subsystems is enhanced [49-51].

END

---

## Round 3 · Author Response

Dear editor,
I am re-submitting the manuscript after revising it according to the reviewer's report in SciPost Physics Core.
Sincerely.

Jeong Ryeol Choi
Department of Nanoengineering, Kyonggi University
Republic of Korea

---

## Round 3 · List of Changes

Phenomenological description of the dissipative effects has been removed.

---

## Editorial Decision

published